

# On the directions and structure of the short-term magnetic variations.

Andrey Khokhlov[1,2,4], Roman Krasnoperov[2], Bogdan Nikolov[2], Julia Nikolova[2], Mikhail Dobrovolsky[2], Dmitry Kudin[2], Valery Petrov[3], and Ivan Belov[2]

[1]Bauman Moscow State Technical University, ul. Baumanskaya 2-ya 5, 105005 Moscow, Russia
[2]Geophysical Center RAS, ul. Molodezhnaya 3, 119296 Moscow, Russia
[3]Pushkov Institute of Terrestrial Magnetism, Ionosphere and Radio Wave Propagation RAS (IZMIRAN), Kaluzhskoe Hwy 4, Troitsk, 108840 Moscow, Russia
[4]Institute of Earthquake Prediction Theory and Mathematical Geophysics RAS, ul. Profsoyuznaya 84/32, 117997 Moscow, Russia

**Correspondence:** A. Khokhlov (fbmotion@gmail.com)

**Abstract.** We study the directional structure of the 1-minute magnetic variations and demonstrate their polarization-like asymmetry: statistically these directions are close to certain 2D plane in 3D space and the orientation of such a plane depends on the observatory location. We describe the method of the magnetic data processing and some partial results, however in this publication we are avoiding the reasonings concerning the physical origins of the detected effect and its consequences for the
observatory data analysis in general.

## 1  Introduction

The geomagnetic field is continuously measured by a network of magnetic observatories, thus, the mathematical and computational technologies for data processing should take into account the actual conditions in which observation of the Earth's magnetic field is performed at these observatories. For instance the effective identification of noise and elimination of its
influence on final data is an important part of the data processing.

A magnetic observatory should be constructed so that only the natural magnetic field is present. In particular, the location is selected so that local magnetic anomalies, be it from geological or artificial origin, must be eliminated and this in principle should lead to the fact that the observable short-term effects expected to be homogeneous. The spatial structure of the magnetic field is therefore well-tested (typically this requires the spatial gradients to be of the order of magnitude $1\ nT\ m^{-1}$), temporal
magnetic field variations must be identical inside the entire observation space. However the study of the short-term variations was mainly reduced to their absolute or component estimations without serious interest to their directional structure.

What is important is that such a study of the directional structure may be extracted not necessarily from the observatory datasets but also from a variometer, which is a magnetometer designed to monitor the temporal variations of magnetic field components relative to a fixed baseline. The numerous magnetic stations therefore give rise to more complete research of the
local specific details of the directional variations. Indeed the installation of the variometer presumes the reference system (the most popular orientations are $H, D, Z$ and $X, Y, Z$) and that the variometer orientation is stable over time.



We address to the directions that result from magnetic variations of the magnetic vector using either variometer or observatory data. Such a direction is a unit 3D-vector and the data is in one-one correspondence with the points of the unit sphere. The plan is to consider some statistical effects using the experimental probability density distribution. Generally speaking an arbitrary collection of data does not allow application of the statistical methods: the core of any statistics assumes that the data

population is the part of some stationary process. For this assumption of stationarity we need some preliminary magnetic data processing since in real conditions the magnetic vector is subjected to daily and long-term secular variations, there are non-perturbed days and magnetic storm distortions, also the sufficient amount of data is necessary to reveal the (initially hidden) statistical effects.

We may use for the 1-minute variations the term "noise" which is by no doubt relative. For instance, temporary signals in

the magnetic field caused by the sources in the ionosphere–magnetosphere are typically considered as noise (especially during ground magnetic survey and interpretation of results), the signals, which have sources closer than a few tens of kilometres, typically considered as noise during the observatory measurements, Santarelli et al. (2014). Of course, there are also some sources of man-made noise such as DC railways, which in the case of appropriate conductivity of the crust can produce a significant effect in the magnetic data at large distances. It is important that these effects in principle differ in their durations,

spectral shapes etc., the resolution of 1 sec and the randomization of the observational periods may reduce the chance that the directional distribution results from the rare fact of the exceptional man-made or natural noise.

With all this precautions it comes out that the experimental probability density of 1-minute variations is not at all directionally isotropic. This is not new, since the registered magnetic variations may result from the induction effects of the electric currents in the crustal layers – in the situation of homogeneity at the local scale one may expect that the directions of magnetic variations

share the horizontal plane. This is not the case however: indeed the 2D plane concentration of directions exists but this plane is in general neither horizontal nor perpendicular to the local magnetic field lines. We checked also the orientation of these planes in various observatories and saw no clear latitudinal or longitudinal dependencies. We may speculate that this is purely local effect but it is not equally obvious from station to station. Also, this effect depends on the strength of variation: the stronger the variation the closer its direction to the preferred 2D-plane.

At present we have no clear physical interpretation. For this we need at least several time-scales to state that this polarization effect is purely local. So we exhibit the approach now and plan to proceed with further research using various scales in time and space.

## 2   Methods

Without the loss of generality we may assume the initial magnetic data to be the sequence $\{\mathbf{B}\}_n$ of the 3D magnetic vectors

$\mathbf{B}_k = \mathbf{B}(t_k)$ where $\delta t = t_{k+1} - t_k$ equals to 1 minute. We aim to construct the population of 1 min magnetic variations $\mathbf{b} = \delta\mathbf{B}$ whatever this means. The real magnetic signal has some memory at a scales larger than 1-minute but we want to exclude this type of correlations in the sequence of variations. Therefore we take into account some other time scale $\Delta t \gg \delta t$ (in our implementation $\Delta t$ is equal to 10 minutes) and perform some preprocessing. The right plot at Fig. 1 shows the almost





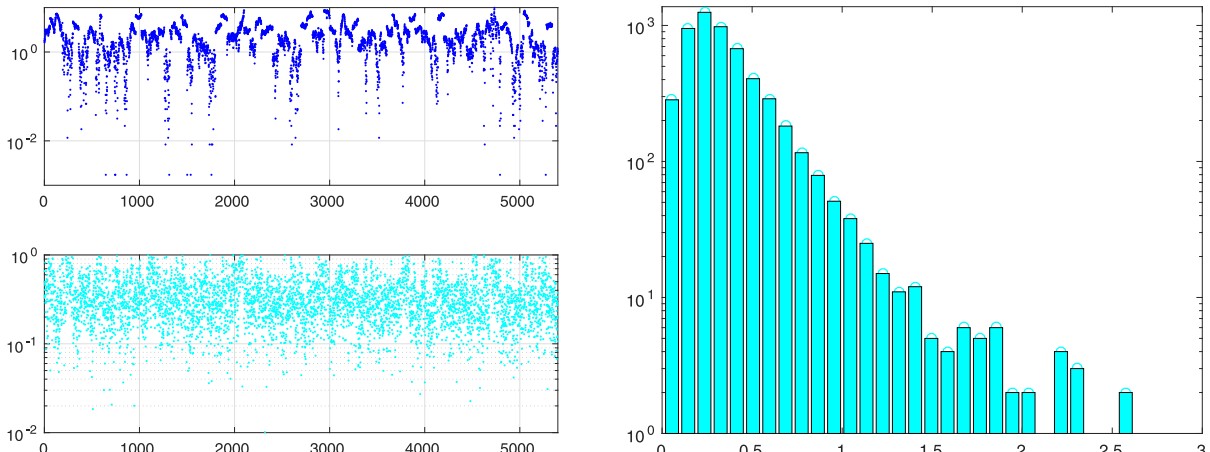

**Figure 1.** Data preprocessing: magnetic data collected from 13.02.2013 to 27.02.2013 for the time interval 12:00–13:00 UTC at Arti (ARS) observatory (location 56.433° N, 58.567° E). Left: intensity of $|\mathbf{B} - \bar{\mathbf{B}}|$ in $nT$, here $\bar{\mathbf{B}}$ is the mean field (top plot) and corresponding absolute values of variations $|\mathbf{b}|$ in $nT$ (bottom plot). Right: histogram of $|\mathbf{b}|$ in the semilogscale.

linear decay of the histogram, this corresponds to almost uniform distribution of $|\mathbf{B}|$ that can be in turn tested directly (such a correspondence is the known fact related to Poisson processes).

## 2.1 Data preprocessing

Let us divide the time into sequential segments $\Delta t$-periods $(t_k, t_k + \Delta t]$ and construct the second order approximation of

$(\mathbf{B}(t_k), \ldots, \mathbf{B}(t_k + \Delta t))$ at each segment – this mimic the $\Delta t$-scale trends of the initial sequence. Now for an arbitrary $t_i \in (t_k, t_k + \Delta t]$ we use the vector value of this approximation $\tilde{\mathbf{B}}(t_i)$ and set $\mathbf{u}_i = \delta\mathbf{B}_i = \mathbf{B}(t_i) - \tilde{\mathbf{B}}(t_i)$.

It can be tested that the sequence $\{\mathbf{b}\}_n$ demonstrates much more stationary behaviour and less correlations — see for instance Fig. 1 — than those in the sequence of the first differences $\{\mathbf{B}_{k+1} - \mathbf{B}_k\}$ (the latter mimic the sequence of time derivatives). Nevertheless at scales $\gg \Delta t$ there persist long-term changes of the absolute values $|\mathbf{b}|$ – this typically reflect the

daily variation of the magnetic activity. The next transformation $\mathbf{b} \mapsto \mathbf{u}$ is the normalization: $\mathbf{u} = \mathbf{b}/|\mathbf{b}|$ therefore we get the set $U$ of unit 3D vectors $\mathbf{u} \in \mathbb{R}^3$. For this we may use magnetic data collected in the fixed UTC time interval (one or many). To illustrate here both approaches we choose the following examples:

1. ARS unperturbed magnetic field. Time interval 12:00–13:00 UTC, data collected from 13.02.2013 to 27.02.2013 at Arti (ARS) observatory (location 56.433° N, 58.567° E),

2. EBR unperturbed magnetic field. Time interval 12:00–18:00 UTC, data collected from 23.05.2015 to 03.06.2015 at Ebro (EBR) observatory (location 40.957° N, 0.333° E),




3. KIV perturbed magnetic field. Data collected 00:00–23:59 01.06.2013 at Kiev (KIV) observatory (location 50.720° N, 30.300° E),

4. SPG perturbed magnetic field. Data collected 00:00–23:59 18.03.2015 at Saint Petersburg (SPG) observatory (location 60.542° N, 29.716° E).

In fact we have considered also many other observatories and data sets. The selected examples are of more or less typical behaviour.

## 2.2 Directional data selection and plots

Actual values for $\mathbf{b}$ that appear during the preprocessing are very small in general and sometimes give zero values for $|\mathbf{b}|$ (because of finite precision of magnetometer) and in that latter case directional unit vector is not even defined. Zeros may also
appear in one or even two components of vector $|\mathbf{b}|$, the corresponding directions then will occupy coordinate planes or axes – but this are exceptional rare situations and we exclude them from $U$ to concentrate our attention on visually obvious features of a distribution over unit sphere $S^2$. The histogram (implemented as polar diagram see Fig. 2) of plane projection of vectors $\mathbf{u}$ onto OXY plane clarify the directional structure of the set $U$, however in incomplete way.

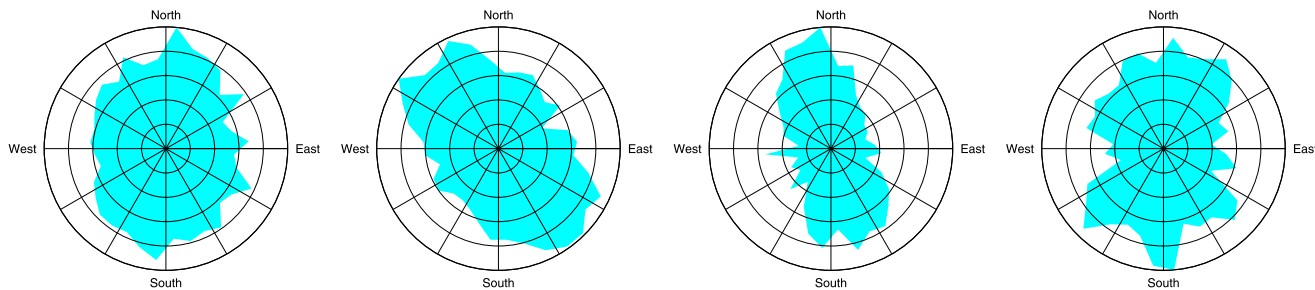

**Figure 2.** Polar diagrams for ARS, EBR, KIV, SPG data (see main text).

In order to consider the 3D structure we may either compute approximative density of points over sphere (see details below)
or use 3D plot to show points at sphere $S^2$; both approaches are shown for the ARS magnetic data at Fig. 3. It is important for the 3D plots to remember that in virtue of the discrete nature of variations (remember the precision of the data!) we sometimes get several different directions as one point at the unit sphere $S^2$.

We may define the average global density of $N$ points over sphere as $\hat{\mu} = N/4\pi$ (where $N$ is the total number of points). For a given latitude interval $(\theta_0, \theta_1)$ in a similar way we define the corresponding density $\mu(\theta_0, \theta_1) = N(\theta_0, \theta_1)/S(\theta_0, \theta_1)$,
here $N(\theta_0, \theta_1)$ is the number of points with latitude $\theta \in (\theta_0, \theta_1)$, $S(\theta_0, \theta_1)$ – the corresponding area. Now we may choose specific latitudes $\{\theta_i\}$ to provide all equal areas $S(\theta_{k-1}, \theta_k)$ and to consider the relative densities $\mu(\theta_{k-1}, \theta_k)/\hat{\mu}$; this may show density irregularities. While the 3D-plot of all points $\mathbf{u} \in U$ may be too cumbersome, the plot of smaller subset may reveal the geometrical pattern of distribution.





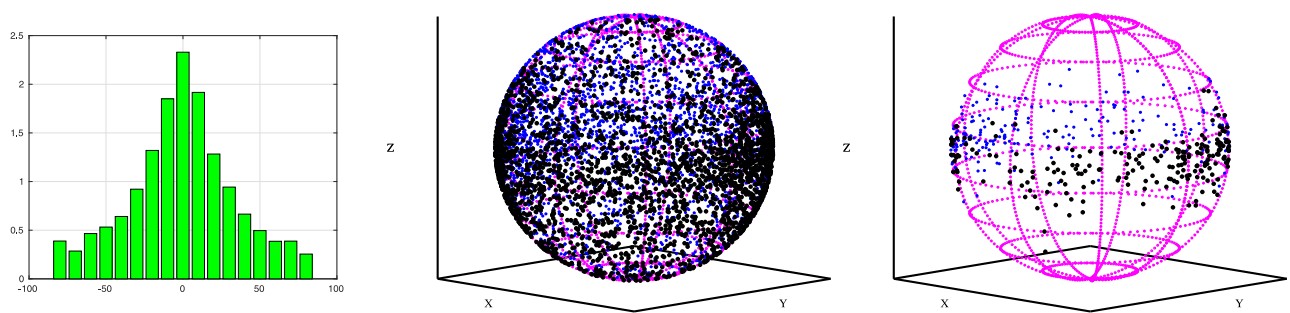

**Figure 3.** The directions $\mathbf{u}$ for the ARS magnetic data (see main text). Left: relative densities for the different latitudes of the unit sphere $S^2$ (see main text), middle: 3D-plot of all directions $\mathbf{u} \in U$, right: shown are only those directions $\mathbf{u} \in U$ that correspond to variations $|\mathbf{b}| > 0.7 \, nT$

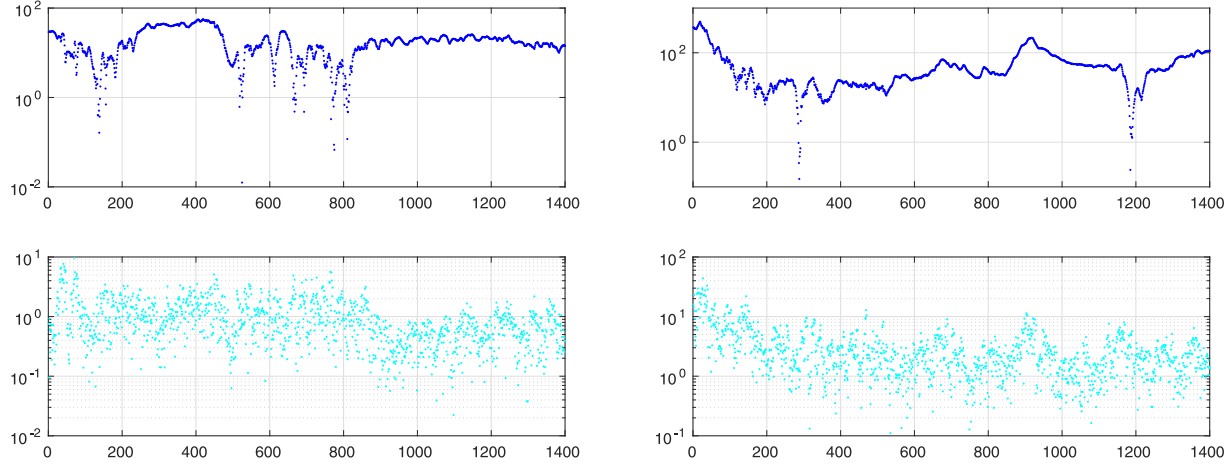

**Figure 4.** Left: intensity $|\mathbf{B} - \bar{\mathbf{B}}|$ in $nT$, $\bar{\mathbf{B}}$ is the mean field (top plot) and corresponding absolute values of variations $|\mathbf{b}|$ in $nT$ (bottom plot) for the KIV data; right: the similar plots for the SPG data.

Now address the KIV and SPG data: in general the typical variations $|\mathbf{b}|$ there are order of magnitude stronger then those of ARS data, see Fig. 4, we also obviously see the inhomogeneous nature of the directional distribution, the projections of the two clusters are shown on Fig. 2.

For the corresponding 3D plots no selection of the strong variations $\mathbf{b}$ is needed: the geometrical structure of density is obvious for both cases. Taking into account 3D-plots of Fig. 5 we may speculate that the clusters of directions $\mathbf{u}$ are located in the vicinity of certain inclined 2D-plane. We may even algorithmically compute the best 2D-plane $Ax + By + Cz = 0$ that approximates the given distribution of $\mathbf{u}$ in each case, indeed we may compute the orientation tensor for all $\mathbf{u}$ directions and





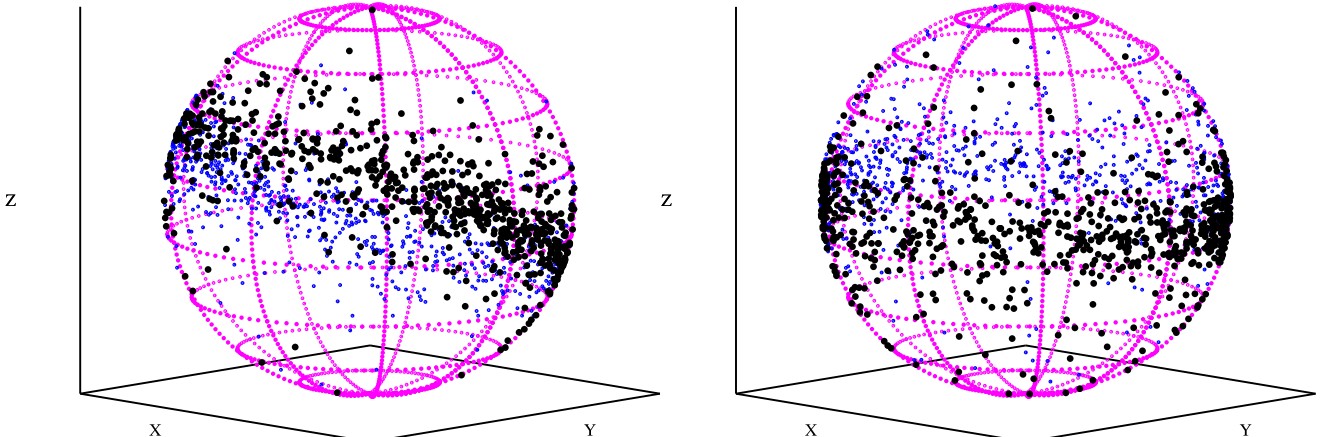

**Figure 5.** Left: 3D plots of all directions $\mathbf{u} \in U$ for the KIV data; right: the similar plot for the SPG data.

derive its eigenvectors and eigenvalues. The smallest eigenvalue defines the 2D plane (that is perpendicular to the corresponding eigenvector $(A, B, C)$), the smaller this eigenvalue the more concentrated $\mathbf{u}$ directions in the vicinity of this plane. These inhomogeneous distributions seem to be the important feature of the magnetometer location which in principle may affect the magnetic data. Let us further name these planes as "polarization planes" for clarity and shortness.

To clarify the relation between the intensities of the variations $\mathbf{b}$ and the densities of the corresponding points at sphere $S^2$ let us consider the mutual scatter plot of the two quantities: $|\mathbf{b}|$ and $\alpha$ where $\alpha$ is the angle (in degrees) between the polarization plane and corresponding $\mathbf{u}$. We see on Fig. 6 the obvious negative correlation between them: the stronger the variation $\mathbf{b}$ the closer its direction to the polarization plane and smaller the angle $\alpha$. This effect was already demonstrated in a different way by Fig. 3, however for KIV data the polarization plane is inclined $30°$ therefore the dependencies of the density from the latitude

are not that obvious.

The polarization plane of EBR data is inclined even more and here we may also use the alternative way to present the relations between $\alpha$ angle and $|\mathbf{b}|$, see Fig. 7. We again have weak variations and therefore polarization plane is less contrast on 3D plot.

### 2.3    Directions and data characteristics

By the nature of preprocessing algorithm the resulting unit vectors $\mathbf{u}_i$ are of different orientations, in real situation we get two big clusters that are oppositely directed. Obviously vectors $\mathbf{u}_k$ are linked to discrete derivatives (first differences) $\{\mathbf{B}_{k+1} - \mathbf{B}_k\}$, however, in virtue of the damping of local trends, we may postulate only qualitatively "the bigger the derivative $|\mathbf{B}_{k+1} - \mathbf{B}_k|$ the bigger the value of $|\mathbf{b}|$". The cluster structure can be revealed in two ways: either make the orthogonal projection of all $\mathbf{u}_i$ onto the polarization plane and then consider plane distribution or compute the usual polar diagram at $OXY$

plane.





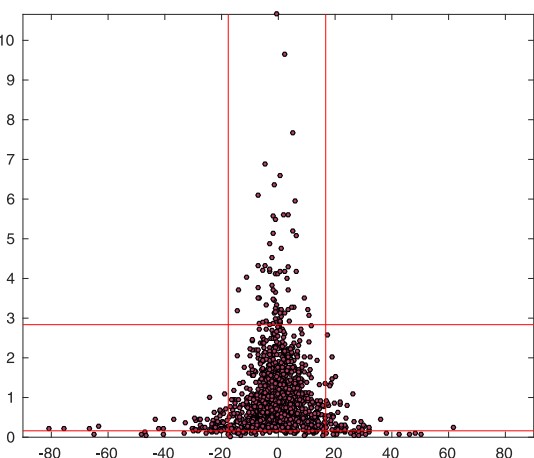 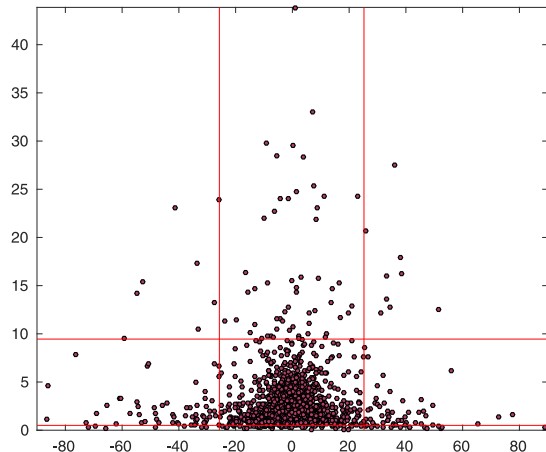

**Figure 6.** Left: shown are the pairs $(\alpha, |\mathbf{b}|)$ (degrees, $nT$) for all directions $\mathbf{u} \in U$ for the KIV data (see main text for the definition of the angle $\alpha$) ; right: the similar plot for the SPG data.

Another type of analysis may clarify the relations between $|\mathbf{b}|$-values and the angles between the corresponding $\mathbf{b}$ and polarization plane (see Fig. 6 ). As a general rule we see here the negative correlation between this two entities: "the bigger the $|\mathbf{b}_k|$ the smaller the angle between $\mathbf{b}$ and the polarization plane". Since the extreme values of field derivatives correspond to large values of $\mathbf{b}$ therefore one may expect the strongest field derivative to be parallel to the polarization plane.

What will happen if we change preprocessing to the simple first-differences algorithm and then normalize the discrete derivatives? Essentially the spherical distribution of directions will be much the same as shown above, however some regular patterns (because of local trends) may appear.

## 3   Results

After the routine cross-check of numerous magnetic observatories we may state the following:

1.  Almost any magnetometer reveals the existence of the polarization plane for the directions of 1-minute variations. How-
        ever, especially if considered the data from only the undisturbed periods of magnetic activity, the polarization effect at
        certain locations can be very weak. In contrast the data from nearabout time of the magnetic storms show the polarization
        effect in much more obvious way.

    2.  The orientation of the polarization plane does not depend on the particular choice of the seasonal data: such a plane
seems to be stable for a given magnetometer at least at the scale from one day to decades.



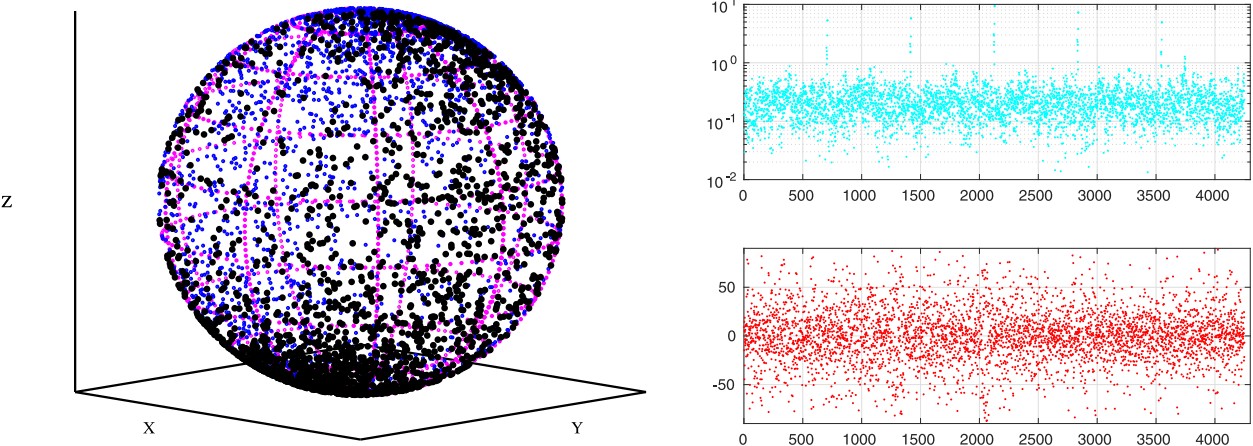

**Figure 7.** Left: 3D plots of all directions $\mathbf{u} \in U$ for the EBR data; right: absolute values of variations $|\mathbf{b}|$ in $nT$ (top) and corresponding $\alpha$ angles in degrees (bottom) for the EBR data (see main text)

3. The orientation of the polarization plane differs from location to location: it need not to be horizontal, sometimes (see Fig. 7 for instance) it is inclined more than $60°$ to the horizontal plane. No clear dependencies between the inclination of the polarization plane and the inclination of the local main magnetic field lines were detected. Also we were yet unable to derive any functional dependency from the geographical locations (latitude and longitude).

4. The strong 1-minute variations of the magnetic field at each observatory demonstrate the clear tendency to stay parallel to the polarization plane. Moreover the strong variations show the structure of two opposite clusters. In other words we may expect the strongest variation to lie within rather limited solid angle of 3D-directions.

## 4   Discussion and conclusions

To stay within the traditional knowledge we may speculate that studied short-term magnetic variations mainly appeared as local induction effects related to the electric currents in the crust. The structure of the conducting media need not be necessarily homogeneous therefore there is no reason to expect that all such electric currents will provide only horizontal magnetic variations. Nevertheless it seems amazing that there exist the stable geometrical structure of the short-term magnetic variations and it is not dependent of the time-scale. We are not yet ready to compare the different time-scales: right now we have considered only the available most frequent magnetic variations.

It is important to remark that this polarization effect appeared obvious only after the non-linear transformation (normalization) of variations. Without this the plane polarization cannot be easily detected by the separate component analysis of the magnetic field.





On the other hand this polarization effect can be easily verified on a wide family of magnetic variometers since we do not need any absolute calibration of the magnetic data. Up to now we used mostly INTERMAGNET observatory data but the network of variometers sometimes is much more dense; the further research will clarify the space scale of the polarization effect using neighbouring magnetic stations.

If only local conditions are responsible for the orientation of the polarization plane we then indeed may have a sort of statistical prediction of the local magnetic activity effects at least at the short time-scale.

*Competing interests.* The authors declare that they have no conflict of interest.

*Acknowledgements.* This work was supported by the Russian Science Foundation grant (project No. 17-77-20034 "Creation of maps of geomagnetic activity characteristics zoning for the territory of the Russian Federation").

The results presented in this paper rely on data collected at magnetic observatories. We thank the national institutes that support them and INTERMAGNET for promoting high standards of magnetic observatory practice (www.intermagnet.org). The presented results are also based on the data collected at Saint Petersburg observatory (SPG). We thank the Geophysical Center of RAS (GC RAS) and Saint Petersburg branch of the Pushkov Institute of Terrestrial Magnetism, Ionosphere and Radiowave Propagation of the Russian Academy of Sciences (IZMIRAN SPb Branch), for supporting its operation and Russian-Ukrainian Geomagnetic Data Center (RUGDC) for making data available

on the Internet. The authors wish to express gratitude to the scientific staff of the Shared Use Facility "Analytical Geomagnetic Data Center" (http://ckp.gcras.ru/).



# References

Santarelli, L., Palangio, P., and De Lauretis, M.: Electromagnetic background noise at L'Aquila Geomagnetic Observatory, Ann. Geophys.-Italy, 57, G0211, doi:10.4401/ag-6299, 2014.