# Peer review of "On the directions and structure of the short-term magnetic variations."

_Annales Geophysicae, 2018_

## Referee Comment (RC1) · Anonymous Referee #1 · 7 Oct 2018

The paper offers visualization tools for geomagnetic field variations applied to data from selected stations at middle latitudes. However, these tools do not provide any quantitative measurements.

Major.

2D polar diagrams simply confirm the well-known fact that the X-component disturbance is somewhat larger than Y-component.

3D diagrams demonstrate the occurrence of an inclined plane with preferred directions of magnetic disturbance vectors. The occurrence of this plane is presented as the main achievement of the paper. In my view, this result is not very significant either as a method or as a scientific result. In fact, methods to determine a plane of specific mag-

netic field fluctuations are widely used in space physics, e.g. the minimum variance analysis (MVA). In contrast to the presented visualization tool, the MVA method determines important quantative parameters, such as orientation angles of normal vector to the plane. The authors are strongly advised to compare their visualization tool with the MVA.

The observed tilt of the "polarization plane" may be caused simply by a lateral gradient of the geoelectric structure of the crust. However, the authors chose not to reveal actual physical reasons of the observed feature. The easiest way to do this is to compare 3D diagrams for two extreme cases: a station above a laterally homogeneous crust and a station with a strong crust gradient (e.g. near the coastline).

The selection of intervals for the study is quite baffling. The authors state that geomagnetically quiet intervals have been selected for investigation. However, magnetograms in Fig. 4 show a disturbance of the B magnitude >100 nT, which is rather high for middle latitudes. At the same time, the authors claim that their effect is more visible for magnetic storms but fail to present any experimental substantiation for this claim.

Minor. 1.16. In fact, variations of geomagnetic field both in magnitude and direction either in space or on the ground were examined in numerous studies.

2.18. Nobody expects magnetic variations to be isotropic above a high conductive Earth's surface.

In summation, from the presented m/s it seems that the described "polarization plane" effect is trivial. The authors failed to demonstrate anything new beyond that. I am convinced that a large team of eight authors can do much better work and produce more rigorous results.

---

## Referee Comment (RC2) · Anonymous Referee #2 · 4 Nov 2018

The article "On the directions and structure of the short-term magnetic variations" presents a new technique to analyse the magnetic variations recorded at ground observatories. This work is based mostly on mathematical consideration and, as the authors point out, it currently does not provide a physical explanation of the results they have found, because its interpretation is not yet completely understood. 4 examples are discussed to illustrate the technique and the difficulties raised during the analysis. They authors affirm that they analysed a much larger set of data coming from additional observatories. Overall, this article presents some weak points that make it unacceptable for publication in its present form. I strongly suggest to revise completely the text and the examples shown, in order to clarify the whole topic. I also suggest strongly to take advantage of the possibility of adding supplementary information that could be useful

fo the readers. A companion document containing additional figures from the observatories under analysis could strengthen the results. It would be extremely beneficial to show, for instance, how the polarisation plane changes during disturbed periods, showing for instance one complete week of daily polarisation spheres.

Another major issue of this manuscript is the lack of references. Only one is provided, that discusses noise at one observatory, while many more would be necessary to support the affirmations that are made and recognise data provides. For instance: a reference to INTERMAGNET (Love, J. J., and A. Chulliat (2013), An international network of magnetic observatories, Eos, Transactions American Geophysical Union, 94(42), 373–374) and to its technical reference manual (Louis, B. S. (Ed.) (2012), INTERMAGNET Technical Reference Manual, ed. 4.6) are necessary, and others pertaining to similar studies.

All figures need to be improved to include axes labels and titles. The information is provided in the caption, but since there is in general very little explanation of the figures, the minimum starting point to understand what is shown, is to provide the labels near the axes. Titles might include at least the observatories code names and the date of analysis shown.

Some specific questions:

I do not understand the affirmation (page 1, line 15) "However the study of the short-term variations was mainly reduced to their absolute or component estimations without serious interest to their directional structure". Without any reference this affirmation is missing a confirmation from the literature.

Page 2 last line: I think a hint to the preprocessing should be added. I understand it is explained later in the manuscript, but a mention to it can be added.

Page 3 first line: Does the plot of figure 1 shows the statistics of |B| of or |b|?

Add a reference to the statistics of Poisson's processes.

I suggest to present the case studies described on page 3 and 4 in a table, and indicate for each how many hours are included in the study.

Does figure 2 include the same amount of data for each diagram?

I think that the whole discussion around "the bigger the |b|' the smaller the angle between b and the polarisation plane", page 6 lines 7-10, Figure 6, page 7 line 1-5, can be shortened. In my understanding this is a consequence of the definition made for the polarisation plane.

Section results: please indicate how many magnetic observatories have been analysed to substantiate the conclusions.

---

## Author Comment (AC1) · 6 Dec 2018

To begin with we must say that we found both comments clear and useful. As a result we decided to withdraw the text in its present form in order to edit the existing text and add the details that may clarify the main target of our research. Indeed, according to that was noticed by the first anonymous reviewer, we also believe that the polarization appeared mainly due to the MT- effect, but initially we tried not to get into this details because in principle the similar effect could be caused also by the poor calibration of the 3-axes of the sensor. Another interesting issue is that the traditional MT sounding using only natural sources is based on rather wide range of frequencies of the variations and the corresponding technology is indeed not that trivial. In contrast our method of recovering the magnetic tipper needs very narrow frequency range and no complicated

data processing. So we are happy that the second anonymous reviewer characterized our research as an attempt of "the new technique" , besides we are completely agree to present " the supplementary information that could be useful to the readers". The amount of this information at present is under consideration, for instance, the accurate comparison of the two approaches of the recovering the magnetic tipper, can be interesting issue in the exploration geophysics. In the present short version we planned only the initial announcement and tried to avoid all that is beyond the short note . Now we'll try to revise and extend our text to make it closer to the practical interests, we agree that our first attempt had many drawbacks.

Thank you again.